# Detection of SARS-CoV-2-Specific Secretory IgA and Neutralizing Antibodies in the Nasal Secretions of Exposed Seronegative Individuals

**DOI:** 10.3390/v16060852

**Published:** 2024-05-27

**Authors:** Jason S. Chwa, Minjun Kim, Yesun Lee, Wesley A. Cheng, Yunho Shin, Jaycee Jumarang, Jeffrey M. Bender, Pia S. Pannaraj

**Affiliations:** 1Keck School of Medicine, University of Southern California, Los Angeles, CA 90033, USA; jchwa@usc.edu; 2Division of Infectious Diseases, Children’s Hospital Los Angeles, Los Angeles, CA 90027, USA; yunhoshin222@g.ucla.edu; 3Division of Infectious Diseases, Department of Pediatrics, University of California San Diego, San Diego, CA 92093, USA; mik075@health.ucsd.edu (M.K.); yel040@health.ucsd.edu (Y.L.); wesleycheng@health.ucsd.edu (W.A.C.); jjumarang@health.ucsd.edu (J.J.); 4Department of Pediatrics, City of Hope Comprehensive Cancer Center, Duarte, CA 91010, USA; jebender@coh.org; 5Division of Infectious Diseases, Rady Children’s Hospital, San Diego, CA 92123, USA

**Keywords:** SARS-CoV-2, COVID-19, seronegative, nasal antibody, secretory IgA, mucosal immunity, exposure

## Abstract

Mucosal immunity may contribute to clearing SARS-CoV-2 infection prior to systemic infection, thereby allowing hosts to remain seronegative. We describe the meaningful detection of SARS-CoV-2-specific nasal mucosal antibodies in a group of exposed-household individuals that evaded systemic infection. Between June 2020 and February 2023, nasopharyngeal swab (NPS) and acute and convalescent blood were collected from individuals exposed to a SARS-CoV-2-confirmed household member. Nasal secretory IgA (SIgA) antibodies targeting the SARS-CoV-2 spike protein were measured using a modified ELISA. Of the 36 exposed individuals without SARS-CoV-2 detected by the RT-PCR of NPS specimens and seronegative for SARS-CoV-2-specific IgG at enrollment and convalescence, 13 (36.1%) had positive SARS-CoV-2-specific SIgA levels detected in the nasal mucosa at enrollment. These individuals had significantly higher nasal SIgA (median 0.52 AU/mL) compared with never-exposed, never-infected controls (0.001 AU/mL) and infected-family participants (0.0002 AU/mL) during the acute visit, respectively (both *p* < 0.001). The nasal SARS-CoV-2-specific SIgA decreased rapidly over two weeks in the exposed seronegative individuals compared to a rise in SIgA in infected-family members. The nasal SARS-CoV-2-specific SIgA may have a protective role in preventing systemic infection.

## 1. Introduction

Individuals exposed to severe acute respiratory syndrome coronavirus 2 (SARS-CoV-2) can experience a variety of outcomes, ranging from asymptomatic infection to severe illness requiring hospitalization. Some individuals remain uninfected according to standard diagnostic tests, never developing PCR positivity or seroconverting [1,2,3,4]. Limited studies have investigated whether mucosal-sequestered antibodies may contribute to the subclinical clearing of SARS-CoV-2 prior to systemic seroconversion [5]. Secretory IgA (SIgA) is present on mucosal surfaces throughout the upper and lower respiratory tract and is protective against respiratory viral diseases [6,7,8]. As SARS-CoV-2 primarily infects the respiratory tract epithelium, SIgA and other neutralizing mucosal antibodies may be critical in providing immunity at the point of entry, eliminating pathogens before the passage through mucosal barriers and identification by the systemic immune system [9,10,11].

The characterization of immunological differences between individuals who evade infection and those who become infected following exposure to SARS-CoV-2 remains pertinent to understanding community transmission, especially as SARS-CoV-2 transmission rates have recently increased with the emergence of novel variants and subvariants [12,13,14]. However, limited data exist on the immune characteristics of respiratory virus exposure without infection. The immune profiles of uninfected exposed individuals may differ from healthy controls or infected individuals. Underscoring a potential protective role played by mucosal antibodies may inform future vaccination strategies, such as the development of intranasal mucosal COVID-19 vaccines. Here, we describe the novel detection of localized SARS-CoV-2-specific nasal mucosal antibodies in a group of exposed-household individuals who evaded systemic infection, confirmed by negative RT-PCR and follow-up serology (hereby collectively termed as “exposed seronegative”).

## 2. Materials and Methods

### 2.1. Study Design and Participants

Participants were enrolled into the Household Exposure and Respiratory Virus Transmission and Immunity Study (HEARTS) following exposure to a household member who tested positive for SARS-CoV-2 by RT-PCR at the Children’s Hospital Los Angeles (CHLA) and nearby community SARS-CoV-2 testing sites in Los Angeles, CA, using a convenience recruitment strategy [9,10,11]. Written, informed consent was obtained from all participants or their parents/legal guardians; assent was obtained from all children ≥7 years old. This study was approved by the Institutional Review Boards at CHLA and the University of California, San Diego.

Demographic information, comorbidities, and exposure history were collected at enrollment via questionnaire. Participants or parents/legal guardians recorded the occurrence and severity of symptoms in a daily symptom diary for 28 days. Participants were considered asymptomatic if they did not report at least one of the following COVID-19-associated symptoms: fever, chills, headache, fatigue, muscle aches, runny nose, congestion, cough, sore throat, shortness of breath, wheeze, altered smell, altered taste, vomiting, diarrhea, or abdominal pain.

### 2.2. Specimen Collection and Processing

Nasopharyngeal swab (NPS) specimens were collected from participants by clinical research staff at enrollment and at subsequent visits every 3–7 days until 2 consecutive negative results were obtained. Blood specimens were collected at enrollment and at 1-month convalescent visits upon the resolution of RT-PCR positivity in the entire household. Blood and NPS specimens in 3 mL universal transport tubes (Becton Dickinson, Franklin Lakes, NJ, USA) were transported to the laboratory for processing within 2 h of collection.

### 2.3. SARS-CoV-2 RT-PCR

RT-PCR for the SARS-CoV-2 N1 and N2 genes was performed on NPS samples as previously published by our group and in accordance with guidelines from the Centers for Disease Control and Prevention [15,16]. Briefly, 50 µL of total nucleic acid eluate was extracted from 200 μL NPS specimens with the QIAamp Viral RNA Mini Kit (QIAGEN, Valencia, CA, USA). RT-PCR reactions were prepared using the 1-Step Taqpath Master Mix (Thermo Fisher, Carlsbad, CA, USA) and primers and probes targeting the SARS-CoV-2 N1 and N2 genes, and Ribonuclease-P (RNP, internal control) (IDT, Coralville, IA, USA). Positivity for SARS-CoV-2 infection was defined by a cycle threshold (Ct) value of <40 for both the N1 and N2 genes. A cutoff Ct of <32 for the Ribonuclease P internal control validated positive PCR results.

### 2.4. Nasal and Serum SARS-CoV-2 Immunoassays

Measurements of nasal SIgA antibodies targeting the SARS-CoV-2 spike protein and total-IgA in NPS were performed using a modified enzyme-linked immunosorbent assay (ELISA). In brief, 96-well plates were either coated with recombinant SARS-CoV-2 spike protein for the SIgA assay or horseradish peroxidase conjugated total-IgA coating antigen for total-IgA (Bethyl Laboratories, A80-102A, Montgomery, TX, USA) and incubated overnight at 4 °C. Plates were washed, blocked, and then incubated with NPS samples diluted at 1:2 for SIgA and 1:500 for total-IgA for two hours at room temperature. The remaining steps were performed with secondary antibodies specific for SIgA (Jackson Laboratories, 115-035-146, West Grove, PA, USA) and total-IgA (Bethyl Laboratories, A80-102P). Assays were performed in duplicates. The optical density values of NPS samples were measured at 490 nm (OD490). Arbitrary units (AU/mL) on a ng/mL scale were calculated from the OD490 values according to the standard curves generated by the known amounts of anti-SARS-CoV-2 spike SIgA and non-specific total-IgA. Immunoassays were performed on 29 control NPS specimens from never-exposed, never-infected individuals (confirmed by negative RT-PCR and serology) collected in early 2020 prior to COVID-19 vaccine availability. Antibody levels greater than the mean AU/mL plus 3 standard deviations of the control samples were used to determine the 6.30 AU/mL and 0.20 AU/mL positive thresholds for SARS-CoV-2-specific unadjusted SIgA and adjusted SIgA, respectively. Linear regression of the standard curves was used to extrapolate each sample’s concentration, which was then multiplied by the respective dilution factor. Nasal SIgA was normalized to nasal total-IgA.

Serum anti-spike receptor-binding domain (RBD) IgG was measured using a previously described ELISA technique [17]. A positive cutoff OD490 value of 0.200 for RBD-specific IgG was calculated from the mean OD490 plus 3 standard deviations (SDs) from 20 archived serum samples collected between 2017 and 2019 prior to the initial SARS-CoV-2 outbreak and from the published protocol.

### 2.5. SARS-CoV-2 Virus Neutralization Assay

Nasal antibody neutralization activity against SARS-CoV-2 was measured using a previously described surrogate virus neutralization assay (GenScript, Piscataway, NJ, USA) [18,19] adapted for NPS samples. In brief, 60 µL of undiluted NPS sample was added to an equivalent volume of horseradish peroxidase conjugated to SARS-CoV-2 RBD protein and incubated for 30 min at room temperature (37 °C). A total of 100 µL of each sample was then added to each well of the angiotensin-converting enzyme 2 coated assay plate and incubated at room temperature for 15 min. The plate was then washed four times, and 100 µL of Tetramethylbenzidine was added to each well. The reaction was quenched using 50 µL of HCl stop solution. Optical density values were measured at 450 nm (OD450). OD450 values were normalized to the negative standard. Lower OD450 values indicate greater neutralization activity. We utilized a 1.0 normalized OD450 positive threshold for neutralization activity.

### 2.6. Statistics

Statistical analyses were performed using R Studio v4.2.0 (Boston, MA, USA). Chi-squared or Fisher’s exact tests were used to compare the proportions of categorical variables. Differences in continuous variables were compared using either Wilcoxon rank-sum tests for unpaired groups or Wilcoxon signed-ranked tests for paired groups, respectively. All tests were two-tailed with *p* < 0.05 considered significant.

## 3. Results

### 3.1. Participants and Nasal Secretory IgA following Acute Exposure

A total of 918 participants from 208 households were enrolled between July 2020 and February 2023. The majority (95.3%) enrolled all household members into the study. SARS-CoV-2 RT-PCR was performed on NPS specimens from 890 (96.9%) participants collected within 9 days (IQR: 6, 14) from the date of the SARS-CoV-2 positivity of the household index case. Of the 372 (41.8%) participants who tested negative, only 36 (9.7%) exposed individuals from 21 (10.1%) households were seronegative for SARS-CoV-2-specific IgG at both enrollment and convalescence (median 92 days (IQR: 78, 97) after exposure) visits. Participant inclusion into our exposed seronegative cohort is summarized in Appendix A. SARS-CoV-2 variants by household included 18 within the ancestral SARS-CoV-2 wave, 1 during the Delta wave, and 2 households in the Omicron wave. The households had an average size of five members including three adults and two children under 18 years of age. 

Among the 36 exposed individuals with SARS-CoV-2 negative acute and convalescent serum, the majority (33/36 [91.7%]) of participants were enrolled during the first year of the COVID-19 pandemic in Los Angeles County (March 2020 to January 2021). Analysis of SARS-CoV-2-specific nasal secretory IgA (SIgA) from collected specimens revealed a spread in mucosal SIgA levels (Figure 1A). Thus, we separated these exposed seronegative (ESN) individuals into two groups with positive and negative nasal SIgA for further analysis (Figure 1B,C). Although 23 (63.9%) ESN household members exhibited negative nasal SIgA levels (ESN-negative), a subset of 13 (36.1%) participants exhibited SARS-CoV-2-specific positive (ESN-positive) mucosal SIgA levels at enrollment. We also evaluated 43 SARS-CoV-2 infected-family members of the exposed seronegative (ESN) individuals with samples collected at the same time. The median timing from exposure to NPS sampling at enrollment (Day 1) was 6 days (IQR: 3, 8) for ESN participants and 5 days (IQR: 3, 7.5) for infected-family members. The demographic characteristics of ESN with negative nasal SIgA, ESN with positive nasal SIgA, and their infected-family members are shown in Table 1. While infected-family members were more likely to exhibit symptoms, no significant differences were seen in age or underlying conditions. 

We performed a subgroup analysis of the ESN-positive individuals compared with the control, ESN-negative, and infected-family members. ESN-positive participants exhibited significantly higher raw, unadjusted SARS-CoV-2-specific nasal mucosal SIgA compared with the control, ESN-negative, and infected-family participants (median AU/mL, ESN-positive [16.5] vs. control [0.01], ESN-negative [0.01], infected-family [0.01], all *p* < 0.001; Figure 1B) at Day 1. When adjusted by nasal mucosal total-IgA, ESN-positive nasal mucosal antibody levels remained higher compared with all other cohorts (median AU/mL, ESN-positive [0.52] vs. control [0.001], ESN-negative [0.001], infected-family [0.0002], all *p* < 0.001; Figure 1C). No differences in unadjusted or adjusted SIgA antibody levels were seen between the control, ESN-negative, and infected-family participants at Day 1.

### 3.2. Exposed Seronegative Nasal Mucosal SIgA-Positive Individuals Exhibit Relatively Transient Antibody Responses Compared with Infected-Family Participants

A second NPS sample was obtained at a median of 8 days (IQR: 7.5, 9.5) after enrollment for ESN-positive individuals and 8 days (IQR: 7, 10) after enrollment for infected-family members. A third NPS sample was collected at a median of 15 days (IQR: 14, 16) for ESN-positive and 15.5 days (IQR: 14, 16.75) after enrollment for infected-family members. Among ESN-positive individuals, nasal SARS-CoV-2-specific SIgA decreased over the 2 weeks compared to Day 1 when evaluated using both unadjusted and adjusted SIgA levels (median unadjusted AU/mL, ESN-positive Day 15 [0.8] vs. Day 8 [5.3] and Day 8 [5.3] vs. Day 1 [16.5], *p* = 0.25 and 0.002, respectively; median adjusted AU/mL, ESN-positive Day 15 [0.02] vs. Day 8 [0.18] and Day 8 [0.18] vs. Day 1 [0.52], *p* = 0.13 and 0.001, respectively, Figure 2A,B). On the other hand, SIgA increased significantly over the 2 weeks among infected individuals (median unadjusted AU/mL, infected-family Day 15 [31.6] vs. Day 8 [23.3] and Day 8 [23.3] vs. Day 1 [0.01], *p* = 0.09 and *p* < 0.001, respectively; median adjusted AU/mL infected-family Day 15 [1.14] vs. Day 8 [0.65] and Day 8 [0.65] vs. Day 1 [0.0002], *p* = 0.09 and *p* < 0.001, respectively). 

In ESN-positive individuals, nasal mucosal antibody neutralization activity was also significantly less robust by Day 8 compared with Day 1 (median normalized OD450, ESN-positive Day 8 [1.00] vs. ESN-positive Day 1 [0.80]; Appendix A). Infected-family members exhibited significantly more robust neutralization activity on Day 8 compared with Day 1 (median normalized OD450, infected-family Day 8 [0.43] vs. infected-family Day 1 [1.00]).

## 4. Discussion

This study describes the detection of nasal mucosal SIgA and neutralizing antibodies in exposed individuals who evaded systemic infection. We found positive nasal mucosal SIgA, and neutralizing antibodies were detected in more than a third of exposed RT-PCR-negative individuals who remained seronegative. The mucosal SIgA levels in ESN-positive individuals were comparable to those elicited in infected-household members one to two weeks after SARS-CoV-2 PCR positivity. Our findings build upon a previous study by Cervia et al., which incidentally found that three healthcare workers (3/19, 15.8%) had detectable mucosal SARS-CoV-2-specific IgA in the absence of seropositivity and PCR positivity [5]. These data demonstrate the importance of nasal mucosal antibodies in providing an initial barrier of protection against SARS-CoV-2 infection and may inform intranasal mucosal vaccine development.

Household members of SARS-CoV-2-infected individuals encompass a unique population of individuals with continual virus exposure over short time periods [20]. While previous studies have mainly involved healthcare workers (HCWs) [21,22,23], our current characterization of individuals with extended SARS-CoV-2 exposure within households is more representative of viral transmission within the general population. This population may experience repeated, prolonged viral exposure within the family unit and lack adequate personal protective equipment (e.g., masks, gloves) and the contact precautions typically observed in clinical settings. Thus, as household members are likely to be subjected to greater viral inoculum, their immune responses may differ, warranting further investigation.

ESN household members with positive nasal mucosal SARS-CoV-2-specific antibodies seemingly evaded infection by mounting a local antibody response at the site of entry before systemic infection could occur. Respiratory viruses such as influenza, rhinovirus, and respiratory syncytial virus have previously been reported to infect nasal mucosal sites and generate mucosal antibodies prior to eliciting a measurable systemic response [24,25,26]. Our data suggest SARS-CoV-2 infection may also lead to robust nasal mucosal antibody responses prior to or without the induction of systemic response, similar to reports in other respiratory virus studies [24,25,26]. Previous studies have reported systemic antibody differences in household children with mild COVID-19 compared with adults despite a similar viral load confirmed by RT-PCR [27], but few studies have characterized localized nasal mucosal antibody responses in SARS-CoV-2-exposed uninfected and infected individuals within the same household [28].

Interestingly, we did not detect significant differences in individual characteristics among ESN-positive, ESN-negative, and infected-family household members, such as age and comorbidity. Notably, we had hypothesized that children would be more likely to avoid infection and/or exhibit a high SIgA antibody response as the high frequency of upper respiratory tract infections associated with early age may prime mucosal innate and adaptive immune responses prior to SARS-CoV-2 infection [29]. Likewise, we expected older individuals would be more likely to have a detectable SARS-CoV-2 infection following exposure as antiviral immune responses generally wane with increasing age [30,31]. Furthermore, we hypothesized individuals with comorbidities to be less likely to evade infection and/or generate a significant SIgA antibody response as a result of compromised host immunity [32]. Ultimately, we were unable to elucidate any potential predictors of the successful evasion of SARS-CoV-2 infection. It is possible that, given our small sample size, the present analysis was not sufficiently powered. 

T cells also contribute significantly to the rapid clearance of SARS-CoV-2 infection [33,34]. Previous studies have proposed that a subclinical viral infection may be rapidly terminated due to an efficient innate and/or adaptive immune response, with the majority demonstrating pre-existing memory T cells capable of cross-recognizing epitope variants as having a significant role [1,2,21,35,36]. Significant demographic predictors of T cell responses enabling infection evasion remain unelucidated, similar to our present findings on the antibody responses of exposed individuals. Nevertheless, it is plausible that pre-existing memory T cells played a role in conferring humoral immunity for our ESN-positive cohort as early T cell proliferation has been observed prior to virus detectability in infection, and T cell receptor clonal expansion typically predates antibody induction following immunological exposure [34,37,38]. 

The durability of nasal antibody responses remains poorly understood. Previous studies have demonstrated nasal antibodies wane after a variable range of 3–9 months following SARS-CoV-2 infection [28,39,40,41]. Furthermore, whether factors such as sex, age, and disease severity impact the strength and longevity of nasal responses remains unclear [5,40]. Whereas our confirmed-infected household members exhibited a heightened mucosal antibody response which continued to increase over two weeks, we observed a transient response in our ESN-positive cohort, which rapidly waned after the first week. Subclinical viral loads and the lack of the persistent antigen stimulation of mucosa-associated lymphoid tissue may be implicated in this transient response.

There are limitations to this study. Infected individuals in our study experienced mild COVID-19 without hospitalization, but this is representative of SARS-CoV-2 infection in the general population. Symptom data reported in patient diaries were self-reported and subject to recall bias if symptoms arose prior to enrollment. Although NPSs were sampled from follow-up visits in enrolled participants, individuals who developed COVID-19 immediately before enrollment or during the follow-up period may have been missed for a positive PCR test. However, non-seroconversion after PCR positivity is rare, and peripheral antibody responses are unlikely to have waned prior to recruitment [42]. Nevertheless, there have been reports of children infected with SARS-CoV-2 diagnosed by PCR who do not seroconvert [43,44], though age-based differences in SARS-CoV-2 seropositivity rates are still debated [45,46]. Furthermore, while some individuals may have been exposed to human coronaviruses (HCoVs) prior to the SARS-CoV-2 outbreak, previously generated antibodies to the S1 subunit found in HCoVs are unlikely to cross-react with SARS-CoV-2 as the S1 subunit shares limited homology with other HCoVs [47,48]. Finally, the antigen used to coat SARS-CoV-2 ELISA plates was specific to ancestral SARS-CoV-2 despite other variants circulating during the study, but the majority of participants in this study were infected with the original SARS-CoV-2 strain.

## 5. Conclusions

Ultimately, with a sufficiently powerful SIgA and neutralizing antibody response, sterilizing immunity may be conferred to the upper respiratory tract, circumventing viral invasion of the lower respiratory tract [28,49]. Prior studies have also reported the association of increased nasal antibody production with decreased viral loads and with more rapid resolution of symptoms or asymptomatic status [28,49]. In our present study, we have identified and described a unique population of SARS-CoV-2-exposed individuals who demonstrated protective mucosal antibody responses despite lack of systemic seroconversion. Future studies should address factors associated with the induction of nasal SIgA response that prevent systemic infection.

## Figures and Tables

**Figure 1 viruses-16-00852-f001:**
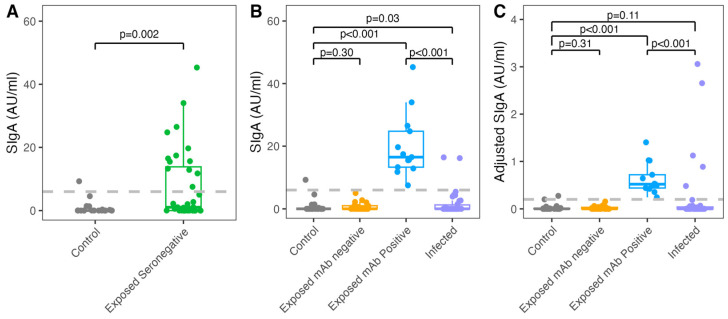
SARS-CoV-2-specific nasal antibody levels in the control, exposed seronegative, and infected participants. (**A**) SARS-CoV-2-specific nasal unadjusted SIgA was compared between the control, seronegative with previous exposure (exposed seronegative), and infected participants. Positive thresholds of 6.30 and 0.20 AU/mL were calculated for unadjusted and adjusted SIgA, respectively, based on the mean plus 3 SD antibody levels of the control samples. ESN participants were stratified by nasal mucosal antibody status using the positive threshold into ESN mucosal antibody (mAb)-positive and ESN mAb-negative groups and compared to the other groups by (**B**) unadjusted SIgA and (**C**) adjusted SIgA normalized by total-IgA. Wilcoxon rank-sum tests were used to determine if median SARS-CoV-2 SIgA values differed significantly. A two-tailed *p* < 0.05 was considered significant.

**Figure 2 viruses-16-00852-f002:**
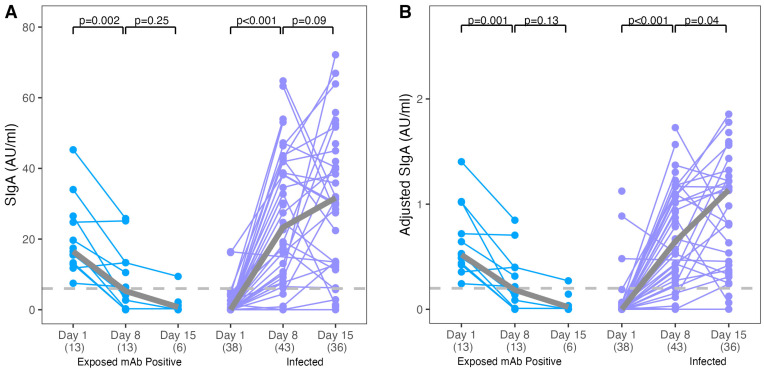
SARS-CoV-2-specific nasal antibody levels at Days 1, 8, and 15 relative to enrollment in exposed seronegative nasal SIgA-positive and infected participants. NPS SARS-CoV-2-specific nasal unadjusted (**A**) and adjusted (**B**) SIgA levels at Day 1 (enrollment), Day 8, and Day 15 are shown for the exposed seronegative and infected participants with positive adjusted SIgA at Day 1, Day 8, and Day 15, respectively. Positive thresholds of 6.30 and 0.20 AU/mL were calculated for unadjusted and adjusted SIgA, respectively, based on the mean plus 3 SD antibody levels of the control samples. Sample sizes (N) are indicated in parentheses. The gray line indicates the trend of the medians for Day 1 and Day 8. Wilcoxon signed-rank tests were used to determine if the median SARS-CoV-2 nasal SIgA values differed significantly. A two-tailed *p* < 0.05 was considered significant.

**Table 1 viruses-16-00852-t001:** Participant demographics and clinical characteristics.

Characteristic	Exposed Seronegative, Nasal SIgA− (*n* = 23) *n* (%)	Exposed Seronegative, Nasal SIgA+ (*n* = 13) *n* (%)	Infected-Family (*n* = 43) *n* (%)	*p*-Value ^a^
**SEX**				0.49
Male	7 (30.4)	6 (46.2)	19 (44.2)	
Female	16 (69.6)	7 (53.8)	24 (55.8)	
**AGE (YEARS)**				0.90
0–17	9 (39.1)	4 (30.8)	16 (37.2)	
18–85	14 (60.9)	9 (69.2)	27 (62.8)	
**RACE**				0.95
Asian	1 (4.3)	1 (7.7)	2 (4.7)	
Black	2 (8.7)	-	3 (7.0)	
White	20 (87.0)	12 (92.3)	37 (86.0)	
Multiple	-	-	1 (2.3)	
**ETHNICITY**				0.81
Hispanic/Latinx	17 (73.9)	11 (84.6)	34 (79.1)	
Non-Hispanic/Latinx	6 (26.1)	2 (15.4)	9 (20.9)	
**COMORBID CONDITION ^b^**	**4 (17.4)**	**4 (30.8)**	**16 (37.2)**	0.58
Asthma/Pulmonary	3	1	6	
Cancer	0	1	1	
Cardiovascular	0	1	3	
Diabetes/Other endocrine	0	0	3	
Immunosuppression or autoimmunity	1	0	0	
Other chronic condition	0	1	3	
**SYMPTOM**				<0.001
Symptomatic	4 (17.4)	3 (23.1)	30 (69.8)	
Asymptomatic	19 (82.6)	10 (76.9)	13 (30.2)	

^a^ Chi-Squared test or Fisher’s exact test. ^b^ At present or past. Comorbid conditions were self-reported and included pre-existing lung, heart, kidney, liver, or neurologic/psychiatric disease; diabetes/endocrine disorder; cancer; pregnancy.

## Data Availability

The data presented in this study are available on request from the corresponding author due to privacy and ethical restrictions.

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
