# Peer review of "Detection of SARS-CoV-2-Specific Secretory IgA and Neutralizing Antibodies in the Nasal Secretions of Exposed Seronegative Individuals"

_viruses, 2024, doi:10.3390/v16060852_

Round 1

Reviewer 1 Report

Comments and Suggestions for Authors

In this manuscript, the authors described a population of SARS-CoV-2 exposed individuals who demonstrated protective mucosal antibody responses despite lack of systemic seroconversion. It is interesting, and helpful for understanding the characteristics of SARS-COV-2. However, 

1. The seronegative or positive was determined by ELISA of anti-RBD IgG, why not determined by neutralization? Sera neutralization antibody is supposed to be more  scientific. 

2. How many participants of the "exposed Seronegative but SIgA positive" group is the first time exposure to COVID-19? Is is possible that these people are convalescent patients, and the increased SIgA is from the previous infection? How can sIgA or neutralized nasal mucosal antibodies be produced in such a short term?

2.The method of the neutralization experiment should be discribed in more detail. 

3.The authors should follow up with the "exposed Seronegative but IgA positive" group in the second or third waves of COVID19. Whether this group of people always show higher IgA than others? If it is true, whether it is related with human genetic diversity?

Author Response

Reviewer 1:

1. The seronegative or positive was determined by ELISA of anti-RBD IgG, why not determined by neutralization? Sera neutralization antibody is supposed to be more  scientific. 

Thank you for this comment. We agree that sera neutralizing antibody has been determined to be the best correlate of protection against symptomatic infection so far. However, because we were looking for evidence of past infection rather than evidence of protection, we used anti-RBD IgG as it is more easily standardized for large-scale screening and surveillance. As seronegativity status of participants were determined prior to first vaccination, a positive anti-RBD IgG would indicate past infection.  In addition, we previously demonstrate a strong correlation between anti-RBD IgG and neutralizing activity in serum of COVID-19 infected individuals (PMID 35632569).

2. How many participants of the "exposed Seronegative but SIgA positive" group is the first time exposure to COVID-19? Is is possible that these people are convalescent patients, and the increased SIgA is from the previous infection? How can sIgA or neutralized nasal mucosal antibodies be produced in such a short term?

Thank for you this comment. While we cannot conclude this with absolute certainty, this was likely the first exposure to COVID-19 for most participants of the ESN-SIgA Positive group. All were seronegative at enrollment without previous report of COVID-19 infection. Of note, 33/36 (91.7%) participants were enrolled during the first year of the COVID-19 pandemic in Los Angeles County (Mar 2020 to Jan 2021). We added this information in the second paragraph of the Results section (Page 3).

The median duration of SARS-CoV-2 exposure to infected symptomatic household members was 9 days at the time of enrollment for all participants. Most antibody responses are seen 7-14 days after exposure. Most ESN members were exposed to multiple infected members, and it is also possible that some may have been exposed to infected asymptomatic household members.

3. The method of the neutralization experiment should be discribed in more detail. 

Thank you for this suggestion. We have provided additional details regarding the methodology of the SARS-CoV-2 Virus Neutralization Assay in Methods, 2.5 SARS-CoV-2 Virus Neutralization Assay (Page 3):

“In brief, 60 µL of undiluted NPS sample was added to an equivalent volume of horseradish peroxidase conjugated to SARS-CoV-2 RBD protein and incubated for 30 min at room temperature (37°C). 100 µL of each sample was then added to each well of the angiotensin-converting enzyme 2 coated assay plate and incubated at room temperature for 15 min. The plate was then washed four times and 100 µL of Tetramethylbenzidine was added to each well. The reaction was quenched using 50 µL of HCl stop solution. Optical density values were measured at 450 nm (OD450). OD450 values were normalized to the negative standard. Lower OD450 values indicate greater neutralization activity. We utilized a 1.0 normalized OD450 positive threshold for neutralization activity.”

4. The authors should follow up with the "exposed Seronegative but IgA positive" group in the second or third waves of COVID19. Whether this group of people always show higher IgA than others? If it is true, whether it is related with human genetic diversity?

This is an excellent suggestion. However, of the 13 individuals in the ESN+ group, 9 received vaccination prior to the second or third waves of COVID-19 and thus did not have comparable immunity status to our original analysis.

Reviewer 2 Report

Comments and Suggestions for Authors

Chwa et al. analyze nasal secretory IgA antibodies in non-infected household contact persons of SARS-CoV-2 infected individuals. They analyze 36 contact persons without detectable virus in swab samples and that did were seronegative for spike-specific IgG at both, enrolment and convalescence. They find in ~1/3 of these seronegative contact persons detectable secretory IgA in nasal swabs at enrolment and speculate that these may be associated with protection.

·         A bit more details should be given in the Methods section to allow easy replication of results, such as sources of reagents, calculation of data, detection limits of assays (surrogate neutralization assay).

·         A Figure, probably as Supplementary Figure, on enrolment of participants, inclusion, exclusion could help to follow the first paragraph of the Results section easier.

·         Axis labels in Figure 1 should be changed to SIgA (AU/ml) and Adjusted SIgA (AU/ml)

·         Title in the third column of Table 1 seems to be incorrect. “Exposed Seronegative, Nasal SIgA- (n=23) n %” should be changed to “Exposed Seronegative, Nasal SIgA+ (n=13) n %”.

·         Table 1 footnote a and b should be changed.

·         Legend for Figure S1 is missing. Figure S1 is generally difficult to understand and for example a cut-off should be added to the graph.

·         Figure 2: n for each time point should be added. As number for day 8 and 15 is sometimes low and not the same individuals, it may be difficult to do a statistic here.

Author Response

Reviewer 2:

Chwa et al. analyze nasal secretory IgA antibodies in non-infected household contact persons of SARS-CoV-2 infected individuals. They analyze 36 contact persons without detectable virus in swab samples and that did were seronegative for spike-specific IgG at both, enrolment and convalescence. They find in ~1/3 of these seronegative contact persons detectable secretory IgA in nasal swabs at enrolment and speculate that these may be associated with protection.

1. A bit more details should be given in the Methods section to allow easy replication of results, such as sources of reagents, calculation of data, detection limits of assays (surrogate neutralization assay).

Thank you for this suggestion.

We have now provided manufacturer information for the universal transport tubes in Methods, 2.2 Specimen Collection and Processing (Page 2) and the reagents used in the RT-PCR protocol in Methods, 2.3 SARS-CoV-2 RT-PCR (Page 2).

In addition, we provided additional details for the neutralization assay in Methods, 2.5 SARS-CoV-2 Virus Neutralization Assay (Page 3), per your suggestion.

2. A Figure, probably as Supplementary Figure, on enrolment of participants, inclusion, exclusion could help to follow the first paragraph of the Results section easier.

Thank you for this suggestion. We have added a Supplementary Figure (Figure S1: Flow Diagram of Exposed Seronegative Cohort.) depicting a flow chart regarding enrollment, inclusion, and exclusion of participants into our exposed seronegative cohort.

3. Axis labels in Figure 1 should be changed to SIgA (AU/ml) and Adjusted SIgA (AU/ml)

Thank you for this suggestion. We have edited the axis labels in Figure 1, as suggested. We have also edited the axis labels in Figure 2 for consistency.

4. Title in the third column of Table 1 seems to be incorrect. “Exposed Seronegative, Nasal SIgA- (n=23) n %” should be changed to “Exposed Seronegative, Nasal SIgA+ (n=13) n %”.

Thank you for catching this error. We have changed the title of the third column of Table 1 accordingly.

5. Table 1 footnote a and b should be changed.

Thank you for catching this error. We have changed Footnotes a and b to correspond with the appropriate sections of Table 1.   

6. Legend for Figure S1 is missing. Figure S1 is generally difficult to understand and for example a cut-off should be added to the graph.

Thank you for this suggestion. We have added a cut-off to the graph.

The legend for Figure S2 (formerly Figure S1) has been uploaded to the submission portal under the “Supplementary caption” field. We have also provided it below:

“Figure S2: SARS-CoV-2-specific neutralization activity at days 1 and 8 relative to enrollment in exposed seronegative nasal SIgA positive and infected participants. NPS SARS-CoV-2-specific neutralization activity is shown for exposed seronegative and infected participants with normalized OD490 at Day 1 and Day 8, respectively. Sample sizes (N) are indicated in parentheses. Wilcoxon signed-rank tests were used to determine if the median SARS-CoV-2 nasal SIgA neutralization activity differed significantly. Two-tailed P < 0.05 was considered significant.”

7. Figure 2: n for each time point should be added. As number for day 8 and 15 is sometimes low and not the same individuals, it may be difficult to do a statistic here.

Thank you for this suggestion. We have included the N for each time point in Figure 2 (as well as Figure S2 [formerly Figure S1]) as suggested.